# L-Band Soil Moisture Retrievals Using Microwave Based Temperature and Filtering. Towards Model-Independent Climate Data Records

**Robin van der Schalie** [1,*], **Mendy van der Vliet** [1], **Nemesio Rodríguez-Fernández** [2], **Wouter A. Dorigo** [3], **Tracy Scanlon** [3], **Wolfgang Preimesberger** [3], **Rémi Madelon** [2] and **Richard A. M. de Jeu** [1]

1. Water & Climate, VanderSat B.V., 2011VK Haarlem, The Netherlands; mvandervliet@vandersat.com (M.v.d.V.); rdejeu@vandersat.com (R.A.M.d.J.)
2. CESBIO (Université Toulouse 3, CNES, CNRS, INRAE, IRD), 31400 Toulouse, France; nemesio.rodriguez-fernandez@univ-tlse3.fr (N.R.-F.); remi.madelon@cesbio.cnes.fr (R.M.)
3. CLIMERS, TU Wien, Department of Geodesy and Geoinformation, 1040 Vienna, Austria; wouter.dorigo@geo.tuwien.ac.at (W.A.D.); tracy.scanlon@geo.tuwien.ac.at (T.S.); wolfgang.preimesberger@geo.tuwien.ac.at (W.P.)
* Correspondence: rvanderschalie@vandersat.com

**Abstract:** The CCI Soil Moisture dataset (CCI SM) is the most extensive climate data record of satellite soil moisture to date. To maximize its function as a climate benchmark, both long-term consistency and (model-) independence are high priorities. Two unique L-band missions integrated into the CCI SM are SMOS and SMAP. However, they lack the high-frequency microwave sensors needed to determine the effective temperature and snow/frozen flagging, and therefore use input from (varying) land surface models. In this study, the impact of replacing this model input by temperature and filtering based on passive microwave observations is evaluated. This is derived from an inter-calibrated dataset (ICTB) based on six passive microwave sensors. Generally, this leads to an expected increase in revisit time, which goes up by about 0.5 days (~15% loss). Only the boreal regions have an increased coverage due to more accurate freeze/thaw detection. The boreal regions become wetter with an increased dynamic range, while the tropics are dryer with decreased dynamics. Other regions show only small differences. The skill was evaluated against ERA5-Land and in situ observations. The average correlation against ERA5-Land increased by 0.05 for SMAP ascending/descending and SMOS ascending, whereas SMOS descending decreased by 0.01. For in situ sensors, the difference is less pronounced, with only a significant change in correlation of 0.04 for SM SMOS ascending. The results indicate that the use of microwave-based input for temperature and filtering is a viable and preferred alternative to the use of land surface models in soil moisture climate data records from passive microwave sensors.

**Keywords:** soil moisture; effective temperature; SMAP; SMOS; LPRM; passive microwave radiometry

## 1. Introduction

The ESA CCI Soil Moisture dataset (CCI SM) [1] is the most extensive climate data record (CDR) of satellite soil moisture (SM) to date, and consists of merged SM retrievals from active and passive microwave satellite sensors all the way back to 1978 [2,3]. In 2010, SM was recognized as an essential climate variable (ECV) by the Global Climate Observing System [4] due to its important role in both land-atmosphere feedbacks on several time scales, and vegetation dynamics on the global scale.

In order to maximize the full development potential of a CDR such as the CCI SM, prioritization of long-term consistency and model independence is needed. If fulfilled, the CDR could be used as a stand-alone climate benchmark, as well as a means to assess state-of-the-art climate and weather models. For example, the Earth System Model evaluation tool [5,6] exploits SM among many other ESA CCI variables to evaluate models in the

Coupled Model Intercomparison Project (CMIP), helping to improve our understanding of the past, present, and future climate.

Two important satellite missions integrated into the CCI SM are the ESA Soil Moisture and Ocean Salinity mission (SMOS) [7] and the NASA Soil Moisture Active Passive mission (SMAP) [8]. These missions are distinguished by their unique L-band (1.4 GHz) radiometers, which are theoretically more suitable for soil moisture retrieval than the previously available higher frequencies such as C-, X-, and Ku-band (6.9 to 18.0 GHz) due to their superior vegetation penetration and deeper sensing depth.

However, these missions lack the onboard sensors needed to determine the effective temperature, an important input parameter for many soil moisture retrieval models, e.g., the Single-Channel Algorithm (SCA) [9], L-Band Microwave Emission of the Biosphere (L-MEB) [10], SMAP Dual-Channel Algorithm (DCA) [11], SMOS-INRA-CESBIO (SMOS-IC) [12], SMOS Neural Networks (SMOS-NN) [13], and the Land Parameter Retrieval Model (LPRM) [14]. Therefore, the retrievals from the current L-band missions make use of temperature and filters derived from global Land Surface Models (LSM) [15]. For a CDR that should function as an independent climate benchmark, this is a disadvantage. However, multi-frequency sensors such as AMSR-2 are capable of retrieving the effective temperature ($T_{mw}$) [16]. These sensors provide filtering for snow/frozen conditions [17] through the optimal use of higher frequency channels, e.g., Ku-, K-, and Ka-bands. This, for example, is already standard in the baseline algorithm for the CCI SM [18], i.e., LPRM, for retrievals from non- L-band missions.

Within this study, the aim is to evaluate the impact of replacing the LSM based input for L-band SM retrievals with one that comes from passive microwave observations, in order to develop an increasingly model-independent CDR. For this, an inter-calibrated brightness temperature dataset (ICTB) is used. The ICTB covers the complete SMOS and SMAP historical record (and further), and consists of AMSR2, AMSR-E, TRMM, GPM, Fengyun-3B, and Fengyun-3D. These satellites are merged together using a minimization function that also penalizes differences in the Microwave Polarization Difference Index (*MPDI*). This allows for a higher level of stability compared to using traditional linear regressions, as the *MPDI* has a pronounced effect on the derivation of the vegetation within LPRM [19]. The main focus will be on the a.m. retrievals, as these are the ones currently used within the CCI, however we will also test the p.m. retrievals for completeness. Studying the effects of using $T_{mw}$ with L-band retrievals will also provide valuable insight for the ESA planned Copernicus Imaging Microwave Radiometer (CIMR) [20] mission, which will carry sensors for L-band, as well as higher frequencies.

The evaluation is structured as follows. Firstly, the results of the inter-calibration activity are presented in order to demonstrate the stability of the ICTB. Secondly, the SM retrievals using microwave based input ($SM_{mw}$) are compared to the SM retrievals using LSM based input ($SM_{mod}$) to define the differences in the characteristics and dynamics of the datasets. The third and final step focuses on changes in skill compared to LSM and in situ data. For this, data were extracted from the European Centre for Medium-Range Weather Forecasts (ECMWF) ERA5-Land Climate Reanalysis [21,22] and the International Soil Moisture Network (ISMN) [23,24]. With a good performance of this new L-band $SM_{mw}$ dataset, a further step towards improving the CCI SM function as a model-independent CDR is made, including a further increase in the consistency of input on temperature and filtering throughout the CCI SM.

## 2. Data

### 2.1. Brightness Temperatures

#### 2.1.1. L-Band Microwave Observations

For this study, L-band observations from SMOS and SMAP are used as a base frequency for the SM retrievals. SMOS and SMAP are unique in the sense that they are the only two major satellite missions dedicated to providing soil moisture observations globally. In addition, both sensors are currently included in the CCI SM, and therefore included in the evaluation.

The SMOS satellite [7], with onboard the Microwave Imaging Radiometer using Aperture Synthesis (MIRAS), has a 2D interferometric radiometer that observes at 1.4 GHz. SMOS is unique in its ability to simultaneously observe in a wide range of incidence angles (0°–65°) and has the longest record of available L-band observations going back to 2010. However, SMOS has issues with Radio Frequency Interference (RFI) [25], mostly over Eurasia. The effective soil temperature used by the official SMOS product is computed using the surface and 50–100 cm depth soil temperature [26] from ECMWF Integrated Forecast System (IFS) model estimates (hereafter, modelled temperature, $T_{mod}$). ECMWF IFS soil temperature for different soil layers, which are spatially resampled to the SMOS EASE grid and temporally interpolated to the SMOS acquisition time, are provided in the AUX_CDFEC auxiliary files.

SMAP [8] was launched in 2015, carrying on board both an L-band conical scanning microwave radiometer at 1.4 GHz and a RADAR. Although the RADAR failed quickly after launch, the radiometer continues to function as planned. Due to the integrated RFI mitigation mechanism, almost all regions have high data quality and, hence, availability. $T_{mod}$, as extracted from the official SMAP product, is generated using the NASA GMAO GEOS-FP model. For SMAP, the $T_{mod}$ is a single value based on the arithmetic mean of two GEOS-FP parameters, i.e., the skin temperature and temperature of the 0–10 cm layer, in order to be representative for the assumed 0–5 cm soil emission layer of L-band.

The L-band data is provided in 25 and 36 km EASE2 grid, and re-gridded to a standard quarter degree grid as used within the CCI SM. For more information on the individual sensors, see Table 1.

**Table 1.** Overview and characteristics of used passive microwave satellite sensors. An asterisk indicates that only V-polarization is available for that band.

| Sensor | Provider | Temporal Coverage (* Still Active) | Bands | Spatial Coverage | Swath Width | Equatorial Crossing Time | Data Level Used |
|---|---|---|---|---|---|---|---|
| Soil Moisture Active Passive Mission (SMAP) | NASA | 04/2015– 12/2020 * | L | Global | 1000 km | Asc: 18:00 Desc: 6:00 | SPL3SMP v7 |
| Soil Moisture and Ocean Salinity Mission (SMOS) | ESA | 01/2010– 12/2020 * | L | Global | 1200 km | Asc: 6:00 Desc: 18:00 | MIR_CDF3T AUX_CDFEC |
| Advanced Microwave Scanning Radiometer for EOS (AMSR-E) on AQUA | JAXA/NASA | 07/2002– 10/2011 | C, X, Ku, K, Ka | Global | 1445 km | Asc: 13:30 Desc: 1:30 | L2A v3 |
| Advanced Microwave Scanning Radiometer 2 (AMSR2) on GCOM-W1 | JAXA/NASA | 05/2012– 12/2020 * | C, X, Ku, K, Ka | Global | 1450 km | Asc: 13:30 Desc: 1:30 | L1R |
| Tropical Rainfall Measuring Mission's (TRMM) Microwave Imager (TMI) | NASA | 01/1998– 12/2013 | X, Ku, K *, Ka | N40º to S40º | 780 or 897 km after orbit boost 8/2001 | Varies (non polar-orbit) | L1C (XCAL) |
| Global Precipitation Measurement (GPM) Microwave Imager (GMI) | NASA | 03/2014– 12/2020 * | X, Ku, K *, Ka | N65º to S65º | 885 km | Varies (non polar-orbit) | L1C (XCAL) |
| Microwave Radiometer Imager (MWRI) on FengYun-3B (FY3B) | CMA/NSMC | 06/2011– 08/2019 | X, Ku, K, Ka | Global | 1400 km | Asc: 13:40 Desc: 1:40 | L1 |
| Microwave Radiometer Imager (MWRI) on FengYun-3B (FY3D) | CMA/NSMC | 01/2019– 12/2020 * | X, Ku, K, Ka | Global | 1400 km | Asc: 14:00 Desc: 2:00 | L1 |

2.1.2. Ku-, K-, and Ka-Band Microwave Observations

For the $T_{mw}$ and filters for snow/frozen conditions using passive microwave observations, there is a need for sufficient overlapping observations between SMOS and SMAP and other sensors; unfortunately, SMAP and SMOS do not simultaneously measure at the higher frequencies needed for temperature retrievals. As no single sensor can provide sufficient observations to solve this independently, a selection is made of multiple satellite sensors that (partly) have overlap with SMOS and SMAP and have similar sensor characteristics. Table 1 provides an overview of the sensors used for the ICTB.

AMSR2 is the main sensor used for SM retrieval from higher frequencies (C-band and up) in the CCI SM, and is therefore chosen as the base for the inter-calibration. Although the frequencies sometimes differ slightly between sensors, they will be calibrated to match the 18.7 GHz (Ku), 23.8 GHz (K), and 36.5 GHz (Ka) channels for both H and V polarizations to align with AMSR2.

AMSR-E, FY3B, and FY3D are three sensors that have very similar crossing times, coverage, and characteristics to the AMSR2 sensor, and are therefore suitable for direct use in the inter-calibration activity. GPM GMI and TRMM TMI are also used, due to their higher revisit time close to the equator and the long time coverage (from 1997 for TRMM). However, due to their varying crossing times, only observations with a maximum difference of 3 h from the 1:30 observations are included. The K-band is only available in V-polarization on TMI and GMI, which impacts the inter-calibration activity for this frequency, as explained in the Methodology Section 3.2. For GPM and TRMM, the dataset prepared by the GPM Inter-satellite Calibration Working Group (XCAL) [27] is used.

*2.2. ERA5-Land*

ERA5-Land [21,22] is an enhanced-resolution rerun of the land component of the ECMWF ERA5 climate reanalysis, and provided through the Climate Data Store (CDS) of the Copernicus Climate Change Services (C3S). ERA5-Land is available with an hourly time step and goes back to 1981.

The volumetric soil water layer 1 (ERA5-Land SM) was used in the skill analysis of the L-band SM datasets and is representative for a 0–7 cm soil depth. Hourly observations closest to the SMOS and SMAP overpasses and within the 2010–2020 time period were selected and regridded from 0.1° to 0.25°. This was achieved by taking the average of all ERA5-Land grid points of which the center falls within the 0.25° box.

*2.3. International Soil Moisture Network*

The International Soil Moisture Network [23,24] is an international effort to collect and maintain a large database of in situ SM observations. This effort is coordinated by the Research Group Climate and Environmental Remote Sensing of the Vienna University of Technology and supported by ESA.

All available SM data were extracted from the ISMN between 2010 and 2020. Sites and sensors were filtered to include only data measured at a maximum sensing depth of 0.12 m, during a minimum time span of 100 days, with a minimum amount of overlapping observations of 50, and a maximum difference between observation times of 2 h for the satellite observation and in situ sensor. Networks that are used in the evaluation can be found in Table 2.

**Table 2.** In situ networks that are extracted from the ISMN and included in the evaluation. An asterisk [*] is used for sites that are only used for SMOS caused by the shorter temporal coverage of SMAP.

| Network Name | | | |
|---|---|---|---|
| AMMA-CATCH * [28–32] | GTK * | MySMNet * | SW-WHU * [33,34] |
| ARM | HOBE * [34] | ORACLE * | SWEX POLAND * [35] |
| AWDN * | HYDROL-NET Perugia * [36] | OZNET * [37,38] | TERENO * [39] |
| BIEBRZA S-1 | HiWATER EHWSN * | PBO H20 [40] | UDC SMOS * [41,42] |
| BNZ-LTER * [43] | ICN * [44] | REMEDHUS * | UMSUOL * |
| COSMOS * [45,46] | IIT KANPUR * | RISMA * [47–49] | USCRN * [50] |
| CTP SMTMN * [51] | IMA CAN1 * [52] | RSMN * | VAS * |
| DAHRA * [53] | IPE | SCAN | VDS |
| FLUXNET-AMERIKAFLUX * | KIHS CMC | SKKU * | WSMN * |
| FMI * | LAB-net [54] | SMOSMANIA [55,56] | iRON [57] |
| FR Aqui * | MAQU * [58] | SNOTEL [59] | |
| GROW | METEROBS * | SOILSCAPE [60,61] | |

## 3. Methodology

### 3.1. Land Parameter Retrieval Model

The main algorithm for passive microwave-based SM retrievals used within the CCI SM is the Land Parameter Retrieval Model (LPRM) [14,15,18]. Similar to most SM retrieval algorithms, LPRM is based on a 0th-order radiative transfer model, i.e., the tau-omega (τ-ω) model [62]. The τ-ω model is used to simulate the soil, vegetation, and atmospheric components in the microwave land surface emission to generate an estimation of the top-of-the-atmosphere TB. Through forward modelling, these results can then be compared to the actual brightness temperatures as observed by the satellite sensor in order to find a solution.

LPRM is unique in the sense that it was the first widely used retrieval algorithm to include an analytical derivation of the Vegetation Optical Depth (VOD) by using the ratio between the V- and H-polarizations [19] instead of using external data sources. This was another step in reducing the effects of external data forcing within the CCI SM product.

The temperature input for SMAP and SMOS LPRM retrievals in the CCI SM (v5 and earlier) are completed using the $T_{mod}$ fields that are provided with the data. For SMOS LPRM the temperature for the 0–7cm layer is used; note that this differs from the official SMOS product.

$T_{mw}$ temperature input is calculated using a straightforward linear relationship [16]:

$$T_{mw} = (0.893 \times TB_{Ka,V} + 44.8) \tag{1}$$

with $TB_{Ka,V}$ as V-polarized Ka-band observations. This is applied to both day-time and night-time ICTB Ka-band observations. For the 6 p.m. L-band retrievals, the mean is taken from the day-time (~1:30 p.m. before) and night-time (~1:30 a.m. after) $T_{mw}$, and only produced when both are available. When looking at the diurnal cycle of temperature (e.g., Figure 6 in [63]), it is expected that, for the L-band emission depth of ~5cm, the difference between 1:30 a.m. and 6 a.m. is below 2 K. For 6 p.m., the difference is expected to be below 3 K for the average of the 1:30 p.m. and 1:30 a.m. the day after. Therefore, despite the time differences, the temperature is assumed to be sufficiently stable following the current method. Skill evaluation of the SM retrievals against in situ observations and ERA5 are used to support these assumptions.

When either $T_{mw}$ or $T_{mod}$ is below or equal to 274.15 °K, retrievals are excluded from the analysis. Secondly, a snow/frozen flag is applied based on the ratio between Ku-, K-, and Ka-band [17]. No extra filtering was necessary for RFI, as these are already removed within the SMAP and SMOS files. The temperature filtering also removes most of the impact of dense precipitation events on brightness temperatures, as this leads to a sharp decline in observed TB. However, separate research is underway to address this in more detail.

### 3.2. Inter-Calibration of High Frequency Observations

Traditionally, inter-calibration of brightness temperatures from different sensors is carried out using linear regression [27] or bias correction [64]. However, the method of least squares, a standard approach in regression analysis, does not lead to a stable solution when applied to the proposed set of AMSR2, AMSR-E, TRMM, GPM, FY3B, and FY3D.

In particular the *MPDI*, a ratio between H/V-Polarization which is used by LPRM for the VOD retrieval, is sensitive to small changes between the datasets left after inter-calibration based on linear least squares. As the unstable *MPDI* affects the ability to properly distinguish between the emission from the vegetation and the soil, this can have pronounced effects on the SM retrieval. Therefore, for this inter-calibration, we propose the following cost function to be minimized in the linear regression instead of a standard least squares approach:

$$Err = \sum RMSE\ TB_H + \sum RMSE\ TB_V + \sum RMSE\ MPDI \tag{2}$$

with:

$$RMSE\ TB_{H/V} = \sqrt{\frac{\sum_{t=1}^{T}\left(TB_{H/V}^{s1} - \left(\alpha \times TB_{H/V}^{s2} + \beta\right)\right)}{T}} \tag{3}$$

$$RMSE\ MPDI = \sqrt{\frac{\sum_{t=1}^{T}\left(\frac{TB_V^{s1} - TB_H^{s1}}{TB_V^{s1} + TB_H^{s1}} - \frac{\left(\alpha \times TB_V^{s2} + \beta\right) - \left(\alpha \times TB_H^{s2} + \beta\right)}{\left(\alpha \times TB_V^{s2} + \beta\right) + \left(\alpha \times TB_H^{s2} + \beta\right)}\right)}{T}} \tag{4}$$

where *TB* is the brightness temperature for either $V/H$ polarization and from the base (*s*1) and calibrated (*s*2) satellite. The $\alpha$ and $\beta$ are the slope and intercept for the linear regression, respectively. The *T* refers to the amount of overlapping observations in time for a single location.

### 3.3. Evaluating Impact

In order to understand the impact of using $T_{mw}$ instead of $T_{mod}$, the corresponding $SM_{mw}$ and $SM_{mod}$ datasets are first compared to one another. This direct comparison will give insight in how and where the characteristics of the SM data change, or where they remain similar. Information is given on the temporal coverage (Cov.), correlation (Corr.), standard deviation (StDev), and mean.

Secondly, in order to achieve a better understanding of how replacing $T_{mod}$ by $T_{mw}$ affects the skill of the datasets, both the $SM_{mw}$ as the $SM_{mod}$ data sets are evaluated using a direct comparison with ERA5 SM and in situ SM from the ISMN. Here, the correlation (Corr.), bias, and unbiased root mean square error (ubRMSE) values are considered. Observations are only used when both $SM_{mw}$ and $SM_{mod}$ have a valid retrieval.

It is important to note that neither ERA5-Land nor the ISMN can be considered as the "truth" and even have different definitions of what they measure or calculate. In situ observations measure an area of about 1 cubic decimeter, while ERA5-Land and remote sensing data cover much larger areas of 100+ square kilometers. However, although an exact fit is not realistic nor desired, Beck et al. (2021) [65] show that the skill obtained between the coarse ERA-Land and local in situ observations can be higher than is currently obtained by the satellite retrievals. Therefore, there is added value in such comparisons.

When interpreting the results, it is beneficial to keep in mind that both an inferior quality of $T_{mw}$ vs. $T_{mod}$, and an unstable ICTB due to high noise or breaks in the $T_{mw}$, could lead to a lower skill of $SM_{mw}$. This infers that the results represent a combination of these two factors.

## 4. Results

### 4.1. Inter-Calibration of Input Data

In Figure 1, the results for the night-time (~1:30 a.m.) Ka-band ICTB are visualized using a time-latitude Hövmoller plot. Ka-band is shown as it has the largest impact on the SM retrievals. Neither the H nor V-polarized TB data show any significant jump between time periods with varying sensors. This stability is also seen with the *MPDI*, where a clear high *MPDI* band can be seen around the Sahara latitudes, while, for the tropical latitudes, very low values can be found due to the dense vegetation coverage.

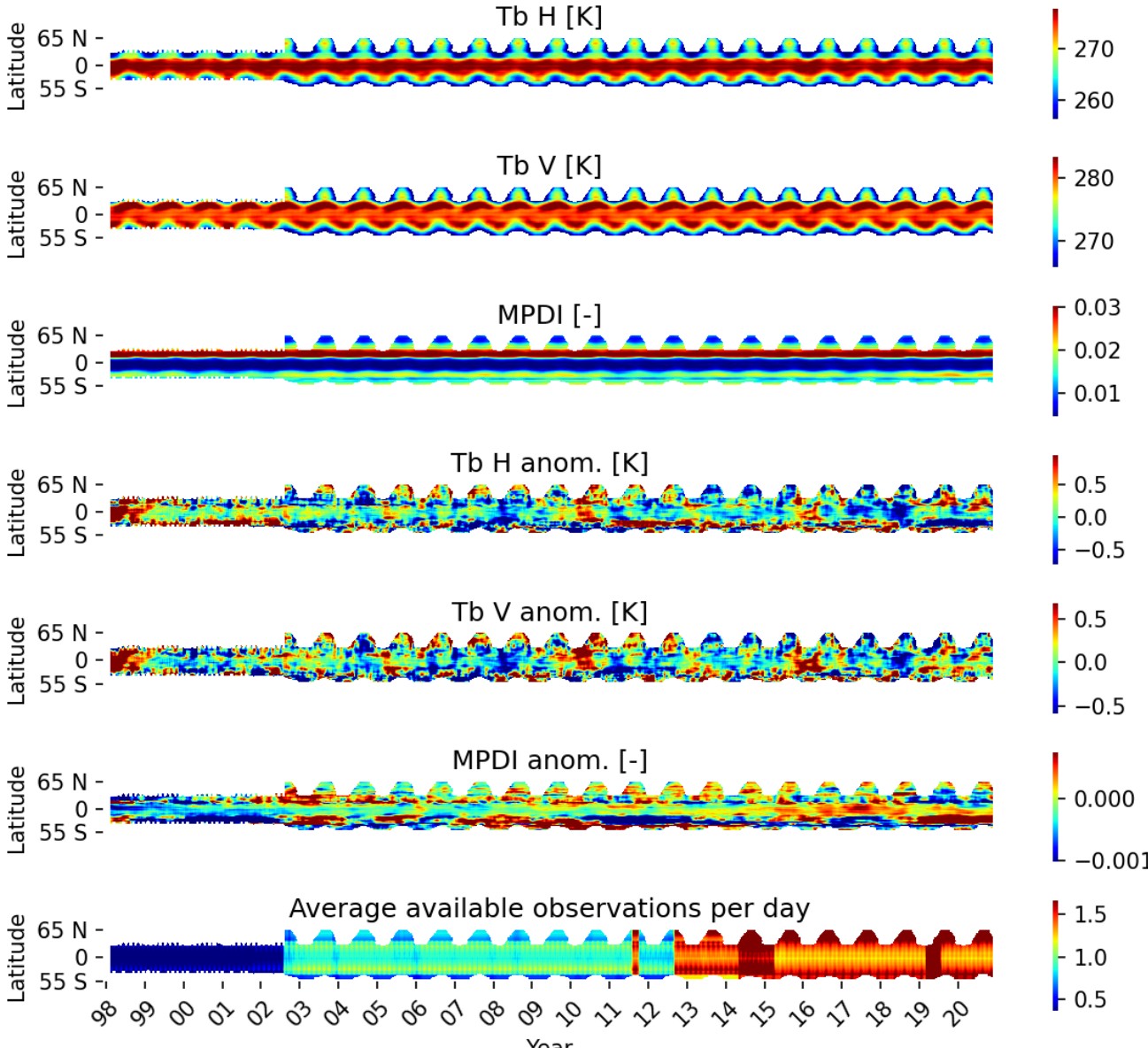

**Figure 1.** Hövmoller plot, a time-latitude plot, of the night-time Ka-band ICTB data that was averaged using a ±30 day moving average. Sub-freezing temperatures have been filtered out. The baseline used for the anomalies is from July 2002 to December 2020 to avoid spatial differences caused by the non-polar orbit of TRMM.

As small inconsistencies might not appear in the wide range of the absolute values, the anomalies were also calculated. It is important to remark that the base period for the anomaly calculation is only from June 2002 onwards, in order to avoid spatial artifacts caused by the reduced TRMM coverage. In the figure, a natural variation of the *TB*

anomalies can be seen, which ranges mostly between −0.75 and 0.75 K, without any structural jumps caused by use of different sensors. For the *MPDI*, this varies between −0.001 and 0.001, with the highest latitudinal differences showing for the southern latitudes, which is caused by the main influence of Australia.

Similar results can be found for the night-time Ku-band ICTB. For K-band however, due to the lack of the H channel on GPM and TRMM, the *MPDI* anomalies clearly differ between the AMSR-E (higher) and AMSR2 (lower) period. The $TB_{K,V}$ anomalies are still within −0.5 and 0.5 K without a break between sensors. The $TB_{K,H}$ anomalies do show some instability, which causes the observed jump in the *MPDI* values. While the day-time ICTB in general performs similarly, there is one clear difference. The varying overpass time of GPM and TRMM leads to a higher *RMSE* in this merging, as compared to the *RMSE* of the night-time merging. For example, for Ka-band, the *RMSE* after merging between TRMM and AMSR2 almost doubles from 0.8 K (night-time) to 1.5 K globally. The *MPDI* is not affected by this.

### 4.2. Inter-Comparison of $SM_{mw}$ to $SM_{mod}$

Figure 2 presents the global maps of the SM SMAP Descending (Desc) inter-comparison, with the results for $SM_{mod}$ in the middle, $SM_{mw}$ on the right, and the difference between the two ($SM_{mw}$–$SM_{mod}$) on the left. The maps show a clear increase in revisit time for the $SM_{mw}$ dataset for all regions except the boreal regions, where a decrease is observed. On average, an increase of 0.46 days is seen.

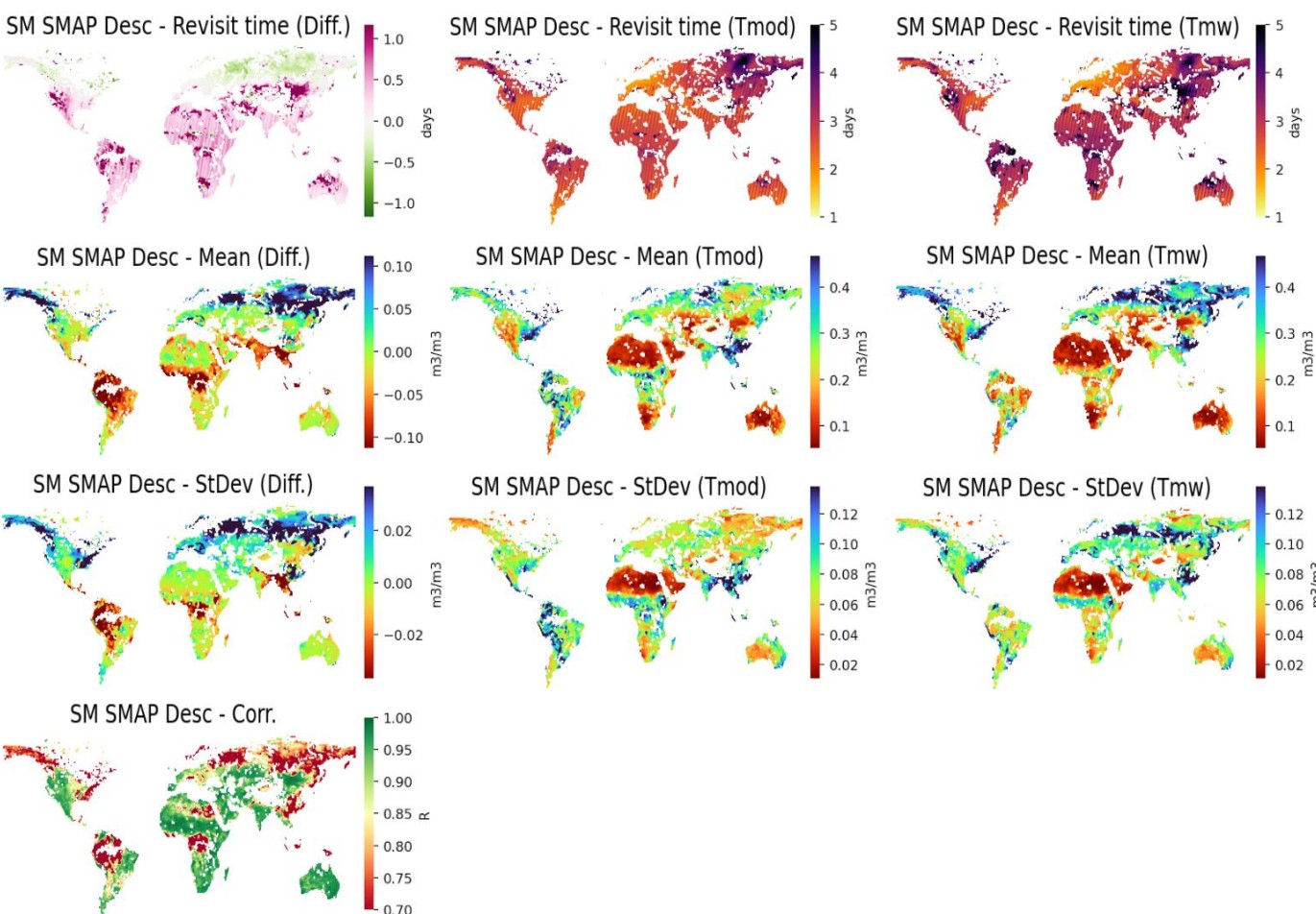

**Figure 2.** SMAP Desc $SM_{mw}$ retrievals compared to $SM_{mod}$.

For the mean and StDev of the SM, the boreal region and the tropics show a strong divergence, while in most regions the results remain similar. The correlation comparison also clearly shows the changed data characteristics over these regions. For the boreal region, the $SM_{mw}$ is wetter, and shows a higher dynamic range, while for the tropical regions the opposite is seen, with the $SM_{mw}$ becoming dryer and less dynamic.

Figure 3 summarizes the information from Figure 2 using cumulative plots of the global data and includes both ascending (Asc) and descending (Desc) datasets of SMAP and SMOS. Spatial patterns as noted before are mostly found in all four datasets. A similar increase in revisit time is seen in all datasets, with an average of 0.57 days for SM SMAP Asc, 0.52 for SM SMOS Asc, and 0.62 for SM SMOS Desc. The change in global mean varies between $-0.02$ for SM SMAP Desc and 0.02 for SM SMOS Desc. For the StDev, on average, no significant difference can be observed. The correlation between SMOS $SM_{mod}$ and $SM_{mw}$ increases much more quickly than for SMAP, however keep in mind that this is caused by a difference in spatial coverage between the two sensors due to RFI in SMOS.

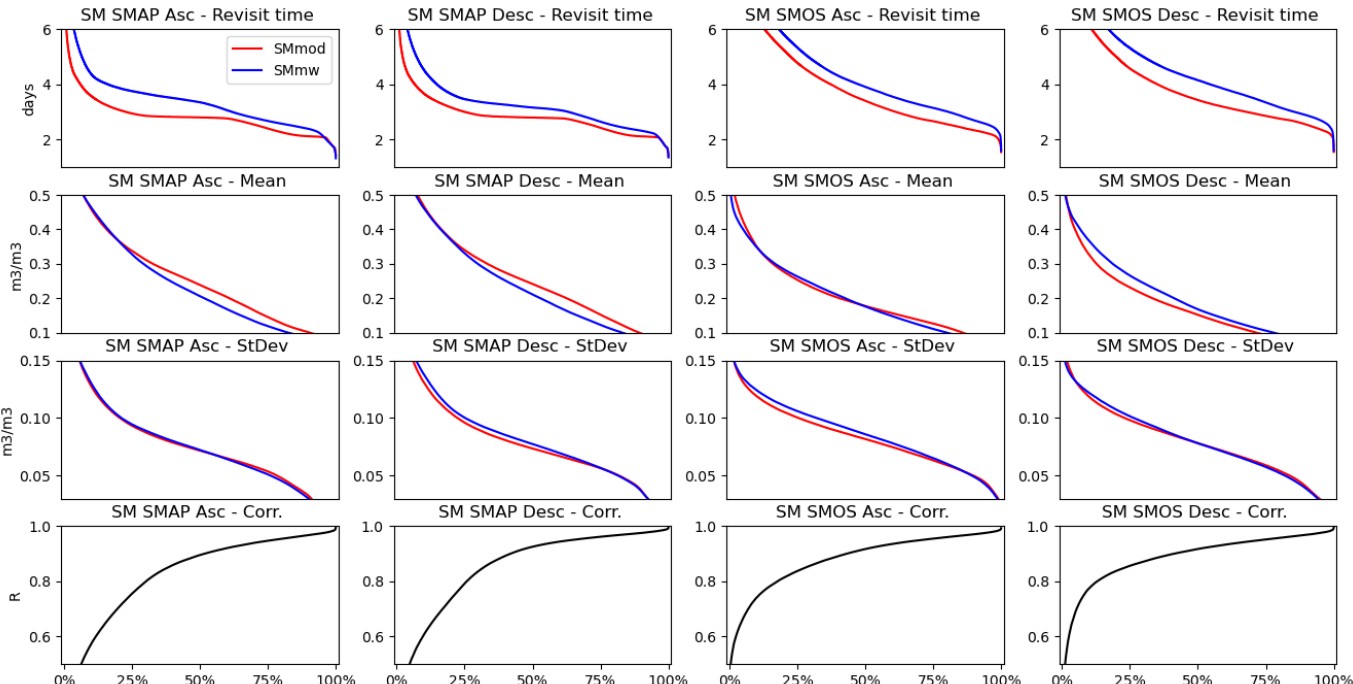

**Figure 3.** Cumulative plots of global SMAP Asc, SMAP Desc, SMOS Asc, and SMOS Desc $SM_{mw}$ and $SM_{mod}$ retrievals for the coverage, bias, standard deviation, and intercorrelation (corr.).

### 4.3. Skill of $SM_{mw}$ Compared to $SM_{mod}$

#### 4.3.1. Based on ERA5-Land

Figure 4 presents the maps of $SM_{mw}$ and $SM_{mod}$ for SM SMAP Desc when evaluating their skill against ERA5. When looking at the correlation, an increase can be detected in all but a few regions, e.g., Mekong delta and Korea. Interesting to note is that the strongest increase can be seen in the boreal regions, where the $SM_{mw}$ often reaches a >0.6 correlation with ERA5. A significant average increase in global correlation of 0.04 is recorded for $SM_{mw}$ compared to $SM_{mod}$.

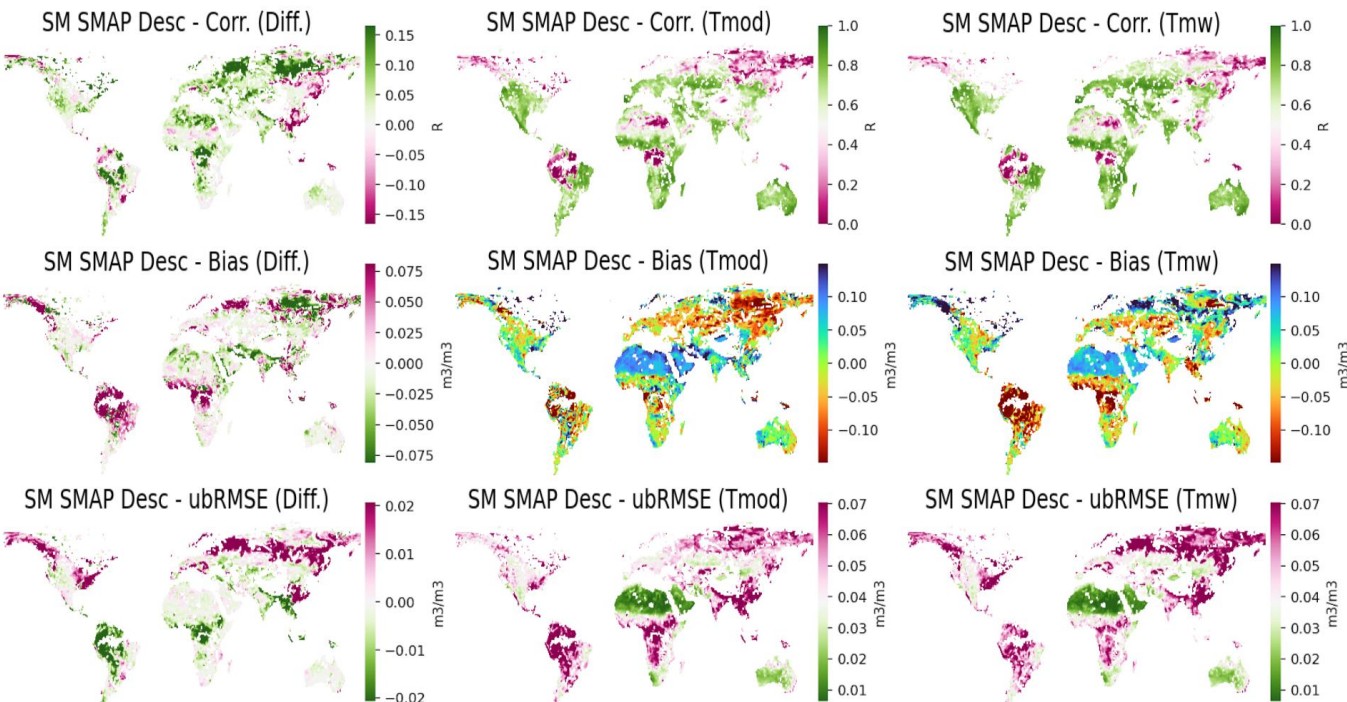

**Figure 4.** SMAP Desc $SM_{mw}$ and $SM_{mod}$ compared to ERA5. Results are shown for correlation (corr.), bias, and ubRMSE.

The bias is mostly neutral, except for the boreal regions where it is a mix of improvement and deterioration, and the tropical regions where it increases. For the ubRMSE, its improved value over the tropical regions aligns with the decrease in the StDev, as shown in the previous section. The increased dynamics (see StDev) in the boreal regions, although having improved correlations, also lead to an increase in the ubRMSE.

Figure 5 summarizes the information from Figure 4 using cumulative plots of the global data and includes both Asc and Desc datasets of SMAP and SMOS. SM SMAP Asc achieves an average improvement in correlation of 0.05, SM SMOS Asc of 0.06, and SM SMOS Desc sees a small decrease of −0.01.

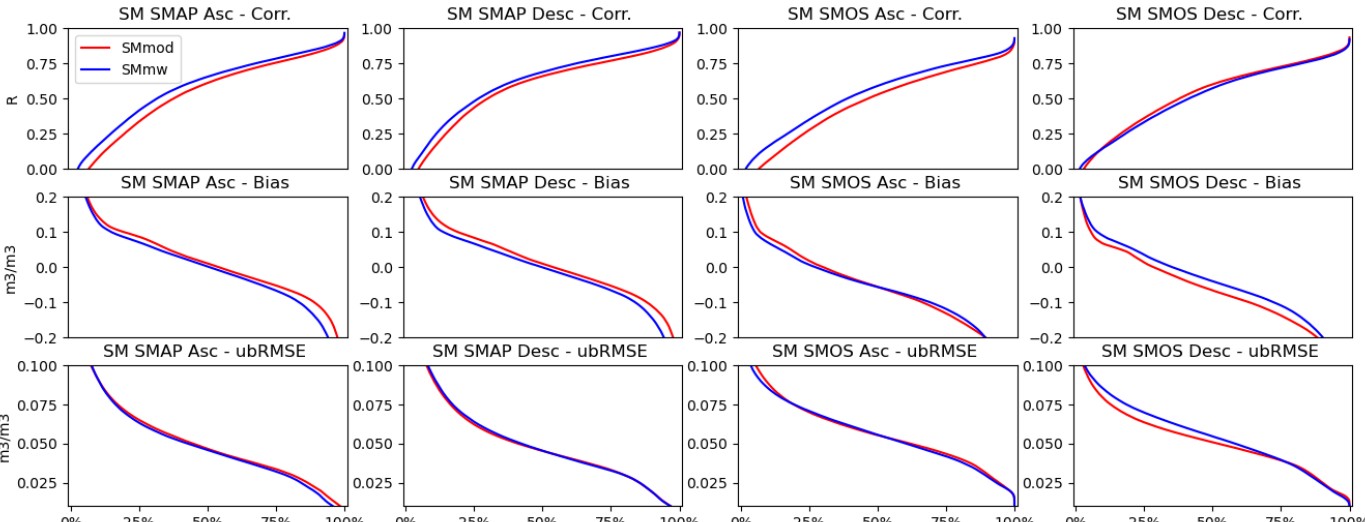

**Figure 5.** Cumulative plots of global SMAP Asc, SMAP Desc, SMOS Asc, and SMOS Desc $SM_{mw}$ and $SM_{mod}$ retrievals compared to ERA5. Results are shown for correlation, bias, and ubRMSE.

### 4.3.2. Based on In Situ Data

Figure 6 shows the cumulative results against all available in situ networks from the ISMN. For SMAP, 224 and 235 in situ sensors are used for the Asc and Desc datasets, respectively. Due to the longer temporal coverage for SMOS, a total of 1045 and 893 in situ sensors were available for the analysis of Asc and Desc, respectively. No significant difference in correlation for SM SMAP Asc and SM SMOS Desc were seen, while SM SMAP Desc and SM SMOS Asc recorded increases of 0.01 and 0.04, respectively. For the bias, only SMOS Desc shows a significant increase of 0.04 m³ m⁻³. The resulting ubRMSE values slightly increase overall, ranging between 0.003 and 0.005 m³ m⁻³.

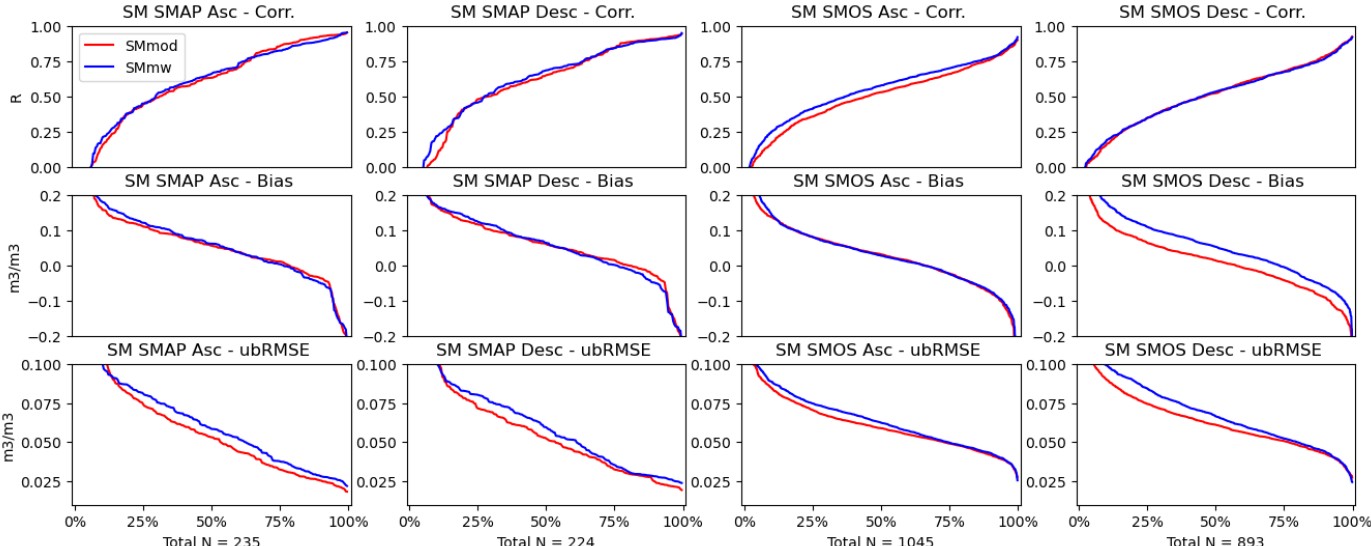

**Figure 6.** Cumulative plots of global SMAP Asc, SMAP Desc, SMOS Asc, and SMOS Desc $SM_{mw}$ and $SM_{mod}$ retrievals compared to in situ data from the ISMN. Results are shown for correlation (Corr.), bias, and ubRMSE.

## 5. Discussion

### 5.1. Inter-Calibration of Input Data

The results from the inter-calibration activity between AMSR2, AMSR-E, GPM, TRMM, FY3B, and FY3D showed a stable consistent dataset of TB, suitable for integration into the L-band soil moisture retrievals. Only the K-band was less stable due to the lack of an H-channel on TRMM and GPM. The $TB_{K,V}$ anomalies, however, are still within $-0.5$ and $0.5$ K, without a visible break between sensors. Therefore, the quality is deemed sufficient for its role in the filtering using the Van der Vliet et al. (2020) [17] method, as it does not use the H-polarization. The $TB_{K,H}$ anomalies are unstable, causing the observed jump in the *MPDI* values. This would only be problematic if it were used in SM retrievals with LPRM as the base frequency for either T or SM. Therefore, when using this methodology for future inter-calibration activities in SM retrievals, e.g., with X-band, this kind of instability is not acceptable.

For the ICTB of day-time observations, the variable overpass time of TRMM and GPM causes higher anomalies in the TB results, while the *MPDI* remains unaffected by this. This expresses itself as higher noise in the temperature input for the LPRM retrievals. However, results in the skill comparisons with ERA5-Land and in situ observations showed this causes no issues, leading to a similar skill for SMOS Desc and an improvement for SMAP Asc.

### 5.2. Inter-Comparison of SM_{mw} to SM_{mod}

When comparing the $SM_{mw}$ to $SM_{mod}$, the most important change is the average increased revisit time of the $SM_{mw}$ globally, except for the boreal regions, where a decrease is even detected. This global increase is caused by the non-full coverage of the ICTB

data, while the decrease over the boreal regions shows the effect of both an almost full coverage ICTB and the snow/frozen filtering. The reader should bear in mind that the shown revisit times are based on successful retrievals, and therefore also include the effect of snow/frozen flagging. This improved flagging effect is what can be seen over the boreal regions. Concerning successful retrievals, there are a few regions, such the Gobi Desert, that have an increase in revisit time higher than just the ICTB coverage effect, caused by an increase in non-converging SM retrievals, i.e., 0 m³ m⁻³, in LPRM with the new temperature.

Although the observations available for the ICTB is above one a day for much of the SMAP period, this can still lead to gaps when there are many simultaneous overpasses. In the future, this could be further improved by extending the sensors within the ICTB with the Special Sensor Microwave Imager Sounders (SSMIS) sensors, FengYun-3C, and, when data would be open and available, WindSat.

The wetter and more dynamic boreal region, and simultaneously a dryer and less dynamic tropical region, is most likely caused by the simplified $T_{mw}$ approach that does not yet include a proper correction of a dynamic atmospheric optical depth. Varying atmospheric water vapor leads to these significant changes on both extreme sides of the spectrum. Keep in mind that for use within the CCI SM dataset, a scaling is applied to the SM during the merging [2,3], which should nullify potential issues with the bias and StDev.

*5.3. Skill of SM$_{mw}$ Compared to SM$_{mod}$*

The results in Section 4.3 show a general better agreement of the SMOS/SMAP SM dynamics with respect to ERA5-Land through the use of $T_{mw}$ instead of $T_{mod}$. Especially worth noting is that even the combined $T_{mw}$, which uses both 1:30 p.m. and 1:30 a.m. (next day) values, leads to a significant increase of 0.05 in correlation globally for the SMAP Asc data. For SMOS Desc, this increase is not seen. The cause for this could be an already similar quality of $T_{mod}$ from ECMWF compared to $T_{mw}$ at approximately 6 p.m. This means the skill of $T_{mod}$ from ECMWF is closer to $T_{mw}$ in the late afternoon as compared to the early morning and that of the SMAP GEOS-FP model. If the cause is in the ICTB, the resulting difference in skill of the SMOS and SMAP afternoon retrievals would be more similar.

The differences as seen in the comparison against in situ are smaller than for ERA5, which is expected, and largely caused by the bulk of the ISMN data coming from the USA. In Sections 4.2 and 4.3.1, this is also a region that does not show much change between the SM$_{mw}$ and SM$_{mod}$. However, they do strengthen the assumption that $T_{mw}$ based on the ICTB can be used as a replacement for $T_{mod}$ for both 6 a.m. and 6 p.m. L-band SM retrievals without a loss in quality, especially as, in the USA, due to the large amount of meteorological and in situ observations available for assimilation, LSM also performs better than average.

With the future launch of CIMR, this skill increase from using $T_{mw}$ is expected to increase further, as then there will be simultaneous observations for L-band, and the higher frequencies needed for $T_{mw}$ and filtering, although the period between 2010 and 2028 will still need to be covered using this, or a similar, method.

## 6. Conclusions

This study presents the use of microwave-based temperature ($T_{mw}$) and filtering input for L-band soil moisture retrievals as a viable alternative to the land surface models currently used as temperature input. Although SMOS and SMAP do not observe in the higher frequencies needed for the $T_{mw}$ and filtering, several satellites with those observations have been active throughout the temporal coverage of both L-band missions, and can be combined into a single inter-calibrated brightness temperature (ICTB) dataset for this purpose.

The ICTB consists of observations from AMSR2, AMSR-E, FY3B, FY3D, GPM, and TRMM, which are merged using a set of linear regressions that use a minimization function

that also penalizes for errors in the *MPDI*. Combined, more than one observation per day are available in most of the L-band period. However, due to the occurrence of simultaneous overpasses, which are also crucial for a correct merging, this does not guarantee a full coverage. The resulting ICTB does not have any breaks in time, and anomalies globally vary naturally between $\pm 0.75$ K.

In order to evaluate the impact of using $T_{mw}$ on the characteristics of the soil moisture retrievals, the $SM_{mw}$ and $SM_{mod}$ were compared to one another. This showed that the revisit time between the SM datasets, increased only by 0.5 days on average. This, surprisingly, also includes the 6 p.m. retrievals that use a combination of 1:30 a.m. and p.m. observations. In general, when comparing the $SM_{mw}$ to $SM_{mod}$, similar mean and StDev values are observed, with correlations of >0.9. There are two major exceptions to this: the boreal regions become wetter with a larger StDev, while the tropics become dryer with a smaller StDev.

The skill of the $SM_{mw}$ was evaluated against two data sources, first the ERA5-Land model and second, the in situ observations from the ISMN. The average correlation against ERA5-Land improved by 0.04, 0.05, and 0.06 for SM SMAP Desc, SM SMAP Asc, and SM SMOS Asc, respectively, with only SM SMOS Desc having a small decrease of $-0.01$. Against in situ observations, the only significant change in correlation was an increase of 0.04 for SM SMOS Asc. The less pronounced results with in situ data can be attributed to the majority of sensors being located in the USA. Further improvements can be expected when addressing the atmospheric effects on the $T_{mw}$ retrieval and by ingesting more satellites, e.g., SSMI and FY3C.

These results show that, for both the 6 a.m. and 6 p.m. L-band retrievals, the use of $T_{mw}$ and microwave-based filtering is not only feasible as an input, but can also contribute to an improvement in skill at the cost of a slightly reduced temporal coverage. The ICTB is sufficiently stable for the retrievals, as can be inferred from the increased correlation, despite the use of TRMM and GPM with their varying overpass times. The ICTB will translate into the CCI SM as a more standardized input for SM retrievals as compared to SM retrievals from higher frequencies with which it is merged, e.g., C-band from AMSR2, and reduce the potential effects of land surface models on the data, strengthening its function as an independent climate data record.

**Author Contributions:** Abbreviation for R.v.d.S., M.v.d.V., N.R.-F., W.A.D., T.S., W.P., R.M. and R.A.M.d.J. The concept and methodology were developed by R.v.d.S., M.v.d.V., and R.A.M.d.J. Intermediate evaluation and support during the progress of the study was provided by M.v.d.V., N.R.-F., T.S., W.P., R.M., and R.A.M.d.J. Writing of the original draft and visualization was primarily completed by R.v.d.S. Supervising was provided by R.A.M.d.J. Project acquisition was primarily achieved by W.A.D. and R.A.M.d.J. All authors have read and agreed to the published version of the manuscript.

**Funding:** This study and the authors were supported by ESA's Climate Change Initiative for Soil Moisture (Contract No. 4000104814/11/I-NB and 4000112226/14/I-NB) and in addition were supported by activities under provision of services to Copernicus Climate Change Service (Contract No. ECMWF/COPERNICUS/2018/C3S_312b_Lot4_EODC/SC2).

**Institutional Review Board Statement:** Not applicable.

**Informed Consent Statement:** Not applicable.

**Data Availability Statement:** The data presented in this study are available on request from the corresponding author. Resulting data from this article is used as an input into the Climate Change for Initiative Soil Moisture processor, which is freely accessible through https://www.esa-soilmoisture-cci.org/data (accessed on 24 June 2021). However, if a reader is specifically interested in one of the datasets presented here, they can be shared for research purposes.

**Conflicts of Interest:** The authors declare no conflict of interest.

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
