# Peer review of "L-Band Soil Moisture Retrievals Using Microwave Based Temperature and Filtering. Towards Model-Independent Climate Data Records"

_remotesensing, doi:10.3390/rs13132480_

Round 1

Reviewer 1 Report

This is a very well written paper reporting an experiment to test the skill improvement of SMAP ascending/descending and SMOS ascending through filtering input of L-band soil moisture retrievals using microwave-based temperature. The results are realistic, good, and very interesting.

However, I think that this manuscript needs the further discussion about the result of SMOS descending, which represents a slight decrement in the performance, makes this manuscript to be constructive. Additionally, equations are not clear due to the coarse resolution of the notation in the text.

I do not have any substantial comments or suggestions for improvements, so I recommend to publish the paper as is and congratulate the authors to this nice piece of work!

Author Response

Dear reviewer 1,

Thank you for your kind words and we are happy to hear that you approve the paper as is. In the revision, we have addressed the two comments that were made. First of all, the equations in the manuscript have been replaced using the Equation Editor in Microsoft Office Word. This should remove the issue with the coarse resolution and notation of the equations. Secondly, to improve the English grammar and style, an English native speaker who is familiar with the scientific content, edited the manuscript. 

Concerning the slight -0.01 correlation of SMOS Descending against ERA5-Land, I’ve added the following sentences to the discussion in order to make the manuscript more constructive:

“For SMOS Desc, this increase is not seen. The cause for this could be an already similar quality of Tmod from ECMWF compared to Tmw  at approximately 6 pm. This means the skill of Tmod from ECMWF is closer to Tmw in the late afternoon as compared to the early morning and that of the SMAP GEOS-FP model. If the cause is in the ICTB, the resulting difference in skill of the SMOS and SMAP afternoon retrievals would be more similar.”

Essentially, this comes down to the varying quality of the input data, both over different models and time of day. We hope these improvements take away the last concerns that you had.

Kind regards,

Robin van der Schalie

Reviewer 2 Report

See attached pdf.

Author Response

Dear reviewer 2,

Thank you for your time and effort put into this thorough review, which really helped us to strengthen the argumentation of the study, leading to an overall improved manuscript. We have addressed your concerns and line-specific comments within the paper and hope we can also clarify these in more detail via this reply.

Primary concern 1: What happens when the Ku-, K-, and Ka-band TB is affected by cloud cover or some other condition.

Answer: In general Ku-, K- and Ka-band TB are little affected by cloud cover. During normal cloud conditions, no adjustments need to be made. Only in case of dense cloud covers due to heavy precipitation events the TB is affected. Generally, this leads to such a drop in Ka-band TB that the retrieved temperature drops to < 1°C, and therefore is filtered out with the temperature filtering. To  make this more robust, a study is currently being done as a follow-up of Van der Vliet et al. (2020), which focuses on improved filtering of (among others) precipitation events. Secondly, another study performed by our team is looking into the integration of an atmospheric module within LPRM that includes an improved atmospheric optical depth integration for the soil moisture retrievals. As compared to clouds, the atmospheric water vapor has a more pronounced influence on the brightness temperatures. In the manuscript we addressed this by improving the filtering description in Section 3.1 and by highlighting the subject in Section 5.2 with the discussion on spatial differences seen between the Tropics and Boreal regions.

Primary concern 2: Is there a hybrid approach that uses Tmod as a fallback? Or is that overpass just completely rejected for the soil moisture retrieval?

Answer: We have decided to exclude these observations for this study. We have the data available that is based on the model-based temperature input, however we find the current loss acceptable. This is done because for the Climate Data Record we want to reduce the impact of the land surface models, in order to function as an independent climate benchmark. Also, because of the merging within the CCI and the availability of multiple other sensors like ASCAT in the combined dataset, the total temporal coverage is less affected than presented here. In section 5.2 we described the outlook of including more sensors to improve the data availability, e.g. WindSat, FengYun-3C and SSMI(S), and even further in the future concerning the upcoming CIMR mission, which will carry onboard a sensor that observes all necessary frequencies.

Primary concern 3: Concerning the assumption between the time difference of observations of the ICTB and L-band.

Answer: We have added the following part to Section 3.1, including the reference to Parinussa et al. (2011):

“When looking at the diurnal cycle of temperature (e.g. Fig. 6 in [30]), it is expected that, for the L-band emission depth of ~5cm, the difference between 1:30 am and 6 am is below 2 K. For 6 pm the difference is expected to be below 3 K for the average of the 1:30 pm and 1:30 am the day after. So, despite the time differences, the temperature is assumed to be sufficiently stable following the current method”.

This gives an indication of the expected quality. Cloud cover is not expected to be a large problem. While frontal passages within this timeframe will affect the temperature, especially when it includes (heavy) rain, this can indeed lead to a degradation of the retrieval skill for that measurement. However, this is also complicated to properly address for the land surface temperature within the LSM. Although this can affect the L-band measurements, the increase in noise on such individual overpasses does not lead to strong degradation of the skill of the retrievals. For example, soil moisture peaks will still be recorded in such an event, even though the temperature might be a few K too high.

Primary concern 4: Context on ERA5-Land and In Situ, which cannot be assumed as the “truth”.

We thank the reviewer for this comment, as we indeed did not address this properly in the text in the previous manuscript. Therefore, we have added the following sentences to Section 3.3 to better put this in perspective:

“It is important to note that ERA5-Land and the ISMN can both not be considered as the “truth” and even have different definitions of what they measure or calculate. In situ observations measure an area of about one cubic decimeter, while ERA5-Land and remote sensing data cover much larger areas of 100+ square kilometers. However, although an exact fit is not realistic nor desired, Beck et al. (2021) [31] showed that the skill obtained between the coarse ERA-Land and local in situ observations can be higher than is currently obtained by the satellite retrievals. Therefore, there is added value in such comparisons. ”

With this sentence, we hope to clarify that an improvement of satellite observations against in situ observations can still be achieved, although no exact 1:1 relation can be expected.

Line specific comments:

  • L038: Instead of just adding the reference to the institute that recognized SM as an ECV, we have written it in full now (see bold text): In 2010, SM was recognized as an essential climate variable (ECV) by the Global Climate Observing System [4]
  • L060: That is correct, however within the CCI the SMAP retrievals are done using the same parameterization as that of SMOS. This paper shows the process of obtaining an LPRM version that is generally applicable to L-band observations.
  • L106: Thank you for noticing this mistake, as it indeed should be surface temperature (not surface soil moisture) in this sentence. We adjusted this accordingly.
  • L115: Both the specific model soil layers were listed, however one in section 2.1.1. and the other in section 3.1. We agree that this should be more clear, therefore both can be found now in section 2.1.1.
  • Tab 1: We have removed Tmod from Table 1 as suggested.
  • L132: Changed “trained” to “calibrated”, to clarify that these are the frequencies that we use as a base for the inter-calibration, while between some sensors there are minor differences in the observed frequency.
  • L167: Slightly adjusted the sentence to clarify that it is about the overlapping observations between the in situ sensor and the satellite observations
  • Tab2: Thank you for pointing out the necessity to include all individual references to the in situ networks used from the ISMN. We have now included them in the manuscript.
  • L192: Thank you for this suggestion, we have removed the file name.
  • L216: This refers to the amount of overlapping observations in time (for a single point spatially, e.g. a certain latitude/longitude combination), we clarified this by rewriting the sentence.
  • L230: The objective is not to increase the SM skill per se, as described in the introduction, the objective is to “Within this study the aim is to evaluate the impact of replacing the LSM based in-put for L-band soil moisture retrievals with one that comes from passive microwave observations, in order to develop an increasingly model-independent CDR”. Although this was described in the final part of the introduction, we have added the bold text to the description of the aim in order to clarify this. Therefore, even a similar quality would be acceptable for us. The increase in quality for most of the data (at the cost of a loss in revisit time), is actually an added benefit of this study.
  • Fig1: Thank for you for spotting this, we have adjusted it as suggested.
  • L273: We have specified it to the “snow/frozen filtering”, this explanation can now be found in the Discussion in Section 5.2.
  • L291: Adjusted to clarify the soil moisture is being scaled.
  • Fig2: This was indeed not correct, we have adjusted the description of the figure to clarify that they are about SMAP Desc.
  • L297: That is correct, first with Figure 2 the difference in revisit time of SMAP Desc is described, and afterwards when discussing Figure 3 we complement this with the results of SMAP Asc, SMOS Asc, and SMOS Desc.

Concerning the final points made by the reviewer, in the revision manuscript the equations have been replaced using the Equation Editor in Microsoft Office Word. This should improve the compatibility of the equations with the rest of the text, as compared to before. Also, we have made an extra check to make sure the Tmod and Tmw properly use a subscript. I hope the improvements take away the last concerns that you had concerning this manuscript.

Kind regards,

Robin van der Schalie
